# Prolonged Secondary Stroke Prevention with Edoxaban: A Long-Term Follow-Up of the SATES Study

**DOI:** 10.3390/brainsci13111541

**Published:** 2023-11-02

**Authors:** Irene Scala, Simone Bellavia, Pier Andrea Rizzo, Jacopo Di Giovanni, Mauro Monforte, Roberta Morosetti, Giacomo Della Marca, Fabio Pilato, Aldobrando Broccolini, Paolo Profice, Giovanni Frisullo

**Affiliations:** 1School of Medicine and Surgery, Catholic University of Sacred Heart, Largo Francesco Vito, 1, 00168 Rome, Italy; irene.scala92@gmail.com (I.S.); bellavia.sim@gmail.com (S.B.); pierandrea.rizzo01@gmail.com (P.A.R.); jacopo.digiovanni@gmail.com (J.D.G.); giacomo.dellamarca@policlinicogemelli.it (G.D.M.); aldobrando.broccolini@policlinicogemelli.it (A.B.); 2Department of Neurosciences, Sense Organs and Thorax, Fondazione Policlinico Universitario Agostino Gemelli IRCCS, 00168 Rome, Italy; mauro.monforte@policlinicogemelli.it (M.M.); roberta.morosetti@policlinicogemelli.it (R.M.); 3Neurology, Neurophysiology and Neurobiology Unit, Department of Medicine, Fondazione Campus Bio-Medico di Roma, 00128 Rome, Italy; f.pilato@unicampus.it; 4UOC Neurologia and Stroke Unit, Mater Olbia Hospital, 07026 Olbia, Italy; paolo.profice@materolbia.com

**Keywords:** NOACs, DOACs, edoxaban, atrial fibrillation, stroke, cardioembolism, NVAF, anticoagulants

## Abstract

Background: Little evidence is available on the long-term efficacy and safety of edoxaban, mainly due to the recent release date. The primary objective of the study was to evaluate the safety of edoxaban, defined by the incidence of major bleedings. We then aimed to evaluate the incidence of thromboembolic events and the persistence of edoxaban therapy in the long-term. Methods: In this observational cohort study, we included ischemic stroke patients enrolled in a previous study to evaluate the safety and efficacy of long-term edoxaban treatment. Data were collected by a trained investigator through a structured telephone interview. Results: Sixty-three subjects (median age 81.0 (73.5–88.0) years, 38.1% male) were included in the study, with a mean follow-up of 4.4 ± 0.7 years (range: 3.2–5.5 years). Only one patient (1.6%, 0.4%/year) presented a major extracranial bleeding, and none had cerebral hemorrhage. Six thromboembolic events occurred in five patients (7.9%): three recurrent strokes, two transient ischemic attacks, and one myocardial infarction (2.2%/year). Over a follow-up period of more than three years, 13 patients discontinued edoxaban (20.6%). **Conclusions:** Edoxaban seems to be effective and safe in the long-term. The persistence rate of edoxaban therapy is optimal after more than three years of treatment.

## 1. Introduction

Ischemic stroke is a leading cause of mortality and long-term disability worldwide [1]. Although several pathogenetic mechanisms may underlie ischemic stroke [2], cardiac embolism plays a major role, since cardioembolic stroke is characterized by a worse prognosis [3] and, in recent years, has become the “leading cause” of stroke [4]. Among numerous heart conditions associated with cerebral embolism, atrial fibrillation (AF) is by far the most frequent, accounting for about 15% of all strokes worldwide [3]. Furthermore, due to the progressive aging of the world population, it is estimated that this prevalence is destined to increase in the coming years, as AF is an age-dependent condition affecting more than 10% of octogenarians [5].

Currently, direct oral anticoagulants (DOACs) are the first-line therapy for non-valvular AF (NVAF) [6], as these anticoagulants have been shown to be safer and more effective for secondary stroke prevention than vitamin K antagonists (VKAs) [6]. The main limitation of DOAC therapy consists of the relatively recent time of introduction on the market, especially for what concerns edoxaban, which was approved by the U.S. Food and Drug Administration (FDA) less than 10 years ago [7], leading to little evidence concerning the long-term follow-up. The lack of data is especially relevant for elderly patients (≥80 years), since, due to the fear of frailty-related bleeding risk, the proportion of patients erroneously treated with underdosed regimens [8] or with inappropriate antithrombotic therapies [9] is particularly high in these subjects.

In a previous study [10], we confirmed that edoxaban 60 mg started within five days of stroke onset was safe in terms of 90-day bleeding risk and occurrence of hemorrhagic transformations of the ischemic brain lesion in the first week after the cerebrovascular event. Based on the above-mentioned premises, we then decided to evaluate the long-term safety and efficacy of edoxaban in the same population, mainly composed of elderly patients. The primary endpoint of this study was to evaluate the long-term (>3 years) safety of edoxaban for secondary stroke prevention in patients with AF, defined through the incidence of major bleedings. We then aimed to evaluate the long-term efficacy of edoxaban in the same population by analyzing the occurrence of stroke recurrence and other thromboembolic events. Finally, we investigated the long-term persistence of edoxaban therapy, reporting a broad description of the causes that led to the suspension or dose reduction of the study drug.

## 2. Materials and Methods

### 2.1. Study Design and Population

In this observational, single-center cohort study, we included the cohort of patients enrolled in a previous study, the “Prospective Observational Study of Safety of Early Treatment with Edoxaban in Patients with Ischemic Stroke and Atrial Fibrillation (SATES Study)” [10]. This study was a prospective, single-center, non-randomized, uncontrolled, interventional study that aimed to assess the safety of early introduction (within 5 days of symptoms onset) of anticoagulant therapy with edoxaban 60 mg in a population of 75 patients with acute ischemic stroke who presented an anterior or posterior Alberta Stroke Program Early CT (ASPECT) score [11,12] ≥ 6 at the baseline CT scan. Details on the original study population and on the design and results of the SATES study can be found in the published manuscript [10].

In the present study, we included all patients previously enrolled in the SATES study for whom we received informed consent, from the patient or from his/her legal representative, to participate in the long-term follow-up. Exclusion criteria included the refusal to participate in the study and inability to obtain recent information on the patient’s clinical status and/or edoxaban intake.

We considered as the primary outcome of safety the occurrence of major bleedings (defined according to the criteria of the International Society on Thrombosis and Hemostasis [13]). Other safety outcomes were the occurrence of minor bleedings, the need for blood transfusions, and hospitalization for any cause. On the other hand, the incidence of thrombotic events (i.e., transient ischemic attacks (TIAs), stroke recurrence, myocardial infarction, pulmonary thromboembolism, and others) was considered the primary outcome of efficacy. We considered death from any cause and cardiovascular events as other outcome measures to assess the efficacy of edoxaban.

The study conformed to the principles of the 1964 Declaration of Helsinki and its later amendments. The research protocol was approved by the ethics committee of Fondazione Policlinico Universitario “A Gemelli” IRCCS—Rome (prot. number 49434/17; study ID:1797). The study was conducted according the Strengthening the Reporting of Observational Studies in Epidemiology (STROBE) guidelines for observational studies [14].

### 2.2. Data Collection

All the data considered in the study were collected by a trained investigator highly experienced in stroke management (I.S.). Information on patients’ clinical conditions at the time of the previous hospitalization for ischemic stroke (i.e., index event) was collected through electronic medical records review. In particular, demographics; clinical and radiological stroke features, such as median National Institute of Health Stroke Scale (NIHSS) and ASPECT score; stroke treatments; comorbidities and cardiovascular risk factors; CHA2DS2-VASC (congestive heart failure, hypertension, age ≥75 years, diabetes mellitus, stroke or transient ischemic attack, vascular disease, age 65 to 74 years, and sex category) [15]; and HAS-BLED (hypertension, abnormal renal/liver function, stroke, bleeding history or predisposition, labile international normalized ratio, age ≥75 years, and drugs/alcohol concomitantly) scores [16] at the time of the index event were recorded for each study participant.

Data on patients’ long-term follow-up were collected by a structured telephone interview with the patient or, in case of diminished capacity, with a family member or a legal representative. All the phone interviews were carried out between 1 and 15 June 2023. The following variables were recorded for each study participant: age at the time of the follow-up, median follow-up duration, incidence of minor and major bleedings, occurrence of thromboembolic events (recurrent stroke, TIA, myocardial infarction, or others), number and causes of hospitalizations, level of functional disability defined through the modified Rankin Scale (mRS), death, and causes of death. Furthermore, data on anticoagulant therapy were recorded for each participant: dose of edoxaban assumed (low vs. standard dose), interruption of anticoagulant therapy, switching to other anticoagulants, specifying the medication assumed, and the causes of edoxaban reduction/suspension.

### 2.3. Statistical Analysis

The sample size of this cross-sectional study was estimated by considering the occurrence of major bleedings as the primary endpoint and by considering an expected annualized rate of 2.75% major bleedings in a DOAC-taking population, as previously reported [17]. Based on these premises and considering a confidence level of 95% and a precision of the estimate of 5%, the minimum number of patients to be included in the study was 41 patients. Gaussian distribution of quantitative variables was assessed through the Shapiro–Wilk test. Quantitative variables were reported as the median and interquartile range (IQR) or as the mean ± standard deviation (SD), as appropriate. Qualitative variables were expressed by absolute and relative percentage frequencies. Comparisons among groups were performed through the Mann–Whitney U-test or Fisher’s exact test, as appropriate. Statistical significance was settled as two-tailed *p* < 0.05. Statistical analyses were performed through the Statistical Package for Social Science (SPSS^®^) software, version 22 (SPSS^®^, Inc., Chicago, IL, USA).

## 3. Results

### 3.1. General Characteristics of the Study Population

From the 75 patients included in the SATES study [10], we enrolled 63 subjects in the present study (median age 81.0 (73.5–88.0) years, age range 52–100 years, 38.1% male), as we were unable to reach 12 patients (Figure 1) due to incomplete or incorrect phone numbers or home addresses. No one declined to participate in the long-term follow-up study.

At the time of the index event, we can appreciate that the median NIHSS at onset was 6.0 (3.0–11.00), while the median ASPECT score was 10.0 (9.0–10.0), suggesting that our population was composed of patients with mild-to-moderate stroke. Thrombolysis was performed in 39.7% of the study population and mechanical thrombectomy in 25.7% of patients. Arterial hypertension was the most represented cardiovascular risk factor, with 44/63 (69.8%), followed by dyslipidemia (15/63, 23.8%). Median CHADSVASC score was 5.0 (4.0–6.0), and median HAS-BLED was 3.0 (3.0–3.0), indicating a population with a moderate–high risk of cardioembolic stroke and high bleeding risk. All the data concerning the index event can be found in Table 1. For details on the original patient population, please refer to our previous article [10].

The mean follow-up time was 4.4 ± 0.7 years (median 4.6 (4.0–4.9) years), ranging from 3.2 to 5.5 years. At the time of the telephone interview, the median mRS score was 3.0 (0.0–4.0) (Table 1).

### 3.2. Edoxaban Persistence

More than three-quarters of our study population maintained edoxaban therapy during the follow-up period (50/63, 79.4%). Among the 13 patients who were not taking edoxaban at the time of the phone interview, two subjects discontinued any type of anticoagulant therapy, while 11 subjects (11/13, 84.6%) switched to other anticoagulants (seven (7/11, 63.6%) to other DOACs, three (3/11, 27.3%) to enoxaparin, and one to VKAs (1/11, 9.1%)). The main reasons for switching anticoagulants were the renal pathology (3/11, 27.3%), followed by the occurrence of major bleedings (1/11, 9.1%), severe anemia requiring blood transfusion (1/11, 9.1%), minor bleedings (1/11, 9.1%), and autoimmune thrombocytopenia (1/11, 9.1%). One patient (1/11, 9.1%) switched to dabigatran for edoxaban unavailability in her home country (i.e., Philippines), while for three subjects (3/11, 27.3%), the reason for anticoagulation switching was not known. For the two patients who interrupted anticoagulant therapy, the reason for this discontinuation was not clarified, but it was not attributable to the occurrence of an outcome event.

In conclusion, less than 5% of our study population (3/63, 4.76%) discontinued edoxaban for the occurrence of a safety outcome and no one for the lack of efficacy.

Detailed information on the adherence to edoxaban therapy is shown in Table 2 and Figure 2.

### 3.3. Low-Dose Edoxaban

Among the 50 patients who maintained edoxaban therapy during the follow-up period, 12 (24.0%) were assuming the low-dose (i.e., 30 mg) regimen at the time of the interview. The median age of low-dose patients was 81.5 (74.8–88.0) years, and nine of them (75%) were octogenarians. The main cause reported by the patients for the reduction to 30 mg of edoxaban was a body weight < 60 kg (8/12, 66.7%), followed by nephropathy (3/12, 25.0%). Please refer to Table 3.

### 3.4. Safety Outcomes

Among our population, only one female patient (1.6%) presented a major bleeding, as she was admitted to the emergency department for a massive epistaxis requiring blood transfusion. No one presented a fatal bleeding or an intracranial hemorrhage. Therefore, the annualized rate of major bleedings in our edoxaban-taking population was 0.4%.

On the other hand, minor bleedings occurred in 18 (28.6%) subjects, mainly consisting of gum (6/63, 9.5%) or fecal (4/63, 6.4%) bleedings. A minority of patients presented with epistaxis (1/63, 1.6%), urinary bleedings (3/63, 4.8%), or massive bruises (3/63, 4.8%). Therefore, the annualized rate of minor bleedings in our edoxaban-taking population was 6.4%.

In addition to the patient who presented a major bleeding, another patient underwent blood transfusion for severe anemia related to a septic state (2/63, 3.2%).

In conclusion, three patients discontinued edoxaban for safety reasons (3/63, 4.8%): one was the woman who presented the major bleeding, while the other two were the patient who underwent blood transfusion for sepsis-related anemia and a woman presenting with recurrent urinary bleedings.

More than half of the patients were hospitalized during the follow-up period (34/63, 54.0%), and 10 out of 34 were hospitalized twice or more (10/63, 15.9%), leading to 44 total hospitalization episodes during the follow-up period. Causes of hospitalization were highly heterogeneous; orthopedic disorders were the most frequent cause (7/44, 15.9%), followed by pneumonia (5/44, 11.4%), medical procedures related to cancer (5/44, 11.4%), and other kinds of infections (4/44, 9.1%). Only three patients (3/44, 6.8%) were hospitalized for thromboembolic events: two patients (2/44, 4.5%) for stroke recurrence and one (1/44, 2.3%) for myocardial infarction (please refer to Figure 3, Panel A). The data on the safety outcomes are summarized in Table 4.

### 3.5. Efficacy Outcomes

Five patients (5/63, 7.9%) presented a thrombotic event during the follow-up period. In detail, two subjects presented three recurrent strokes (2/63, 3.2%), another two (2/63, 3.2%) a TIA, and only a female patient a myocardial infarction (1/63, 1.6%). Therefore, the annualized rate of thromboembolic events was 2.2%, and the annualized rate of stroke recurrence was 1.1%.

Thirteen subjects died (13/63, 20.6%); cancer was the most frequent cause of death (3/13, 23.1%), followed jointly by heart failure (2/13, 15.4%), cardiac arrest of unknown cause (2/13, 15.4%), and sepsis (2/13, 15.4%). Of note, only one patient (1/13, 7.7%) died from stroke recurrence. To summarize, cardiovascular accidents were the cause of death for five patients (5/63, 7.9%) (details on the causes of death are available in Figure 3, Panel B). Details on this point are available in Table 4.

### 3.6. Comparisons between Low-Dose and Standard-Dose Edoxaban Therapy

Considering the 50 patients who maintained edoxaban therapy at the time of the long-term follow-up, we performed a statistical comparison between patients taking edoxaban at the standard dose (60 mg) and those assuming low-dose edoxaban therapy (30 mg).

We observed that patients taking low-dose edoxaban presented a higher median age than those assuming edoxaban 60 mg (standard dose: 80.0 (71.5–84.8) years vs. low dose: 86.5 (83.5–89.0) years; *p* = 0.010). Female patients tended to more frequently assume a low dose of edoxaban than men (standard dose: 19 (50%) vs. low dose: 10 (83.3%); *p* = 0.051).

Concerning the analysis of outcome events, we found that only TIAs tended to be more frequent in the low-dose group, without reaching statistical significance (standard dose: 0 (0%) vs. low dose: 2 (16.7%); *p* = 0.054). All the other outcome events did not differ significantly between groups. For details, please refer to Table 5.

### 3.7. Comparisons between Short- and Long-Term Efficacy and Safety Data

Comparing the efficacy and safety data from the 3-month follow-up of our previous study [10] and the long-term follow-up in our cohort of stroke patients, we observed higher annualized rates of both major (annualized rate of short-term vs. long term follow-up: 10.7% vs. 0.4%) and minor bleedings (annualized rate of short-term vs. long term follow-up: 58.7% vs. 6.4%) during the short-term period than the long-term one. On the other hand, the annualized rate of thromboembolic events was slightly higher in the long-term follow-up cohort of patients. For details, please refer to Table 6.

## 4. Discussion

In this study, we found that edoxaban therapy seems to be safe and effective in the long term (>3 years of follow-up). In particular, only one patient presented a major extracranial bleeding (1.6%, annualized rate 0.4%), which led to edoxaban discontinuation, and no subjects presented an intracranial hemorrhage or a fatal bleeding. Furthermore, two patients presented three recurrent strokes, accounting for an annualized incidence rate of 1.1%. On the other hand, the overall incidence of thromboembolic events was low, involving 7.9% of our population, with an annualized rate of 2.2%. Finally, the persistence rate of edoxaban therapy was markedly high, considering that only 20.6% of elderly subjects discontinued the drug over a mean period of 4.4 years.

Considering the safety data, we found an occurrence of major bleedings in our patient population lower than the one observed in the edoxaban pivotal trial, the ENGAGE-TIMI 48 [17], which reported an annualized rate of 2.75% with high-dose edoxaban and 1.61% with low-dose edoxaban. Similarly, an observational study analyzing an Asian patient population in primary stroke prevention with edoxaban for NVAF found an annualized major bleeding rate of 2.32% [18]. In another observational study including AF patients from Europe, Japan, and other Asian countries, the major bleeding rate was 1.12%/year [19]. The incidence of minor bleedings in our patient population (annualized rate of 6.4%) fell within the range observed in literature, ranging from an annualized rate of 3.52% (for the low dose) [17] to 7.23% [20].

Concerning the efficacy data, the occurrence of thromboembolic events in our study is slightly higher than the ones reported by Giugliano et al. [17] and the ETNA-AF Program [19] but lower than that reported in Asian patients [18].

The small sample size of our study and the lack of a direct comparison with a subgroup of patients taking other anticoagulants certainly does not allow us to demonstrate the efficacy and safety of edoxaban compared to other DOACs or VKAs, although we can conclude that, in our population composed of elderly patients in secondary stroke prevention, long-term treatment with edoxaban has proved safe, effective, and well tolerated.

Indeed, the old age of our patients, with a median age of 81.0 (73.5–88.0) years, further strengthens the results of the study, since advanced age is associated with a significant increase in both ischemic and hemorrhagic events following anticoagulant therapy [21,22]. We can then hypothesize that the low rates of both safety and efficacy outcome events occurring in our patients may be linked to the selection and the multidisciplinary management of acute stroke patients, since the prescription of NOACs by physicians specialized in the management of atrial fibrillation has been associated with a greater appropriateness of anticoagulant prescription in previous studies [23,24] and, consequently, with a lower rate of adverse events [25].

In our study, only 20.6% of patients discontinued edoxaban therapy over a period of more than three years, showing an optimal degree of treatment persistence. This result can confirm previous evidence suggesting that once-daily DOAC administration increases adherence to anticoagulant therapy, especially in elderly populations [26]. Additionally, recent studies have reported a higher degree of drug persistence in patients taking edoxaban, compared to those taking other DOACs, although the long-term persistence rate was lower when compared to our patients [27,28]. We can therefore assume that the high persistence rate of edoxaban in our population could be related to the subtype of patients enrolled in the study, which was a selected population of subjects who had been previously extensively informed about the benefits and risks of edoxaban in the context of a clinical study [10,26].

The main strength of our study lies in the long-term follow-up of our edoxaban-treated patient population, mainly composed of elderly patients. On the other hand, our study has several limitations: the small sample size; the collection of long-term follow-up data through a telephone interview, as patients or their legal representatives were not always able to accurately report subjects’ clinical information, leading to missing data; and the dissimilar follow-up period among subjects, which ranged from 3.2 to 5.5 years.

## 5. Conclusions

Our study suggests that edoxaban seems to be an effective, safe, and well-tolerated drug in elderly patients with NVAF and previous ischemic stroke.

However, due to many limitations, further prospective, multicenter, larger-sample studies are needed to strengthen the evidence emerging from our study.

## Figures and Tables

**Figure 1 brainsci-13-01541-f001:**
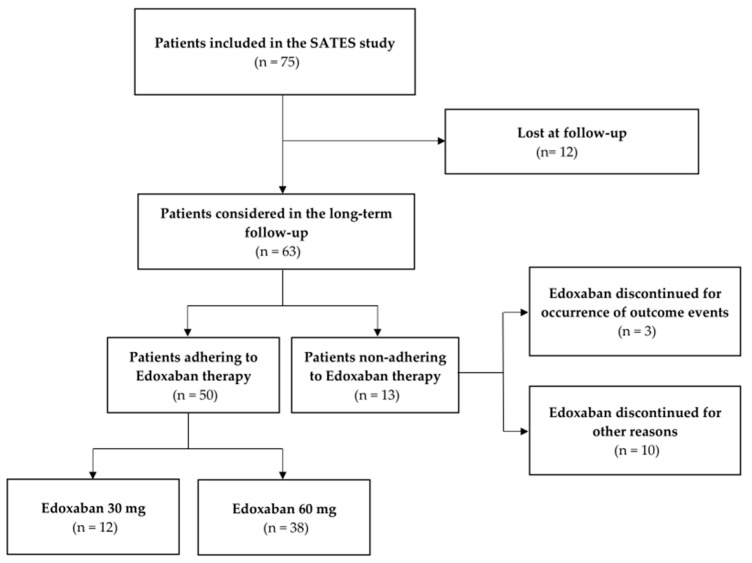
The flow diagram of the enrollment process. Abbreviations: SATES, Prospective Observational Study of Safety of Early Treatment with Edoxaban in Patients with Ischemic Stroke and Atrial Fibrillation.

**Figure 2 brainsci-13-01541-f002:**
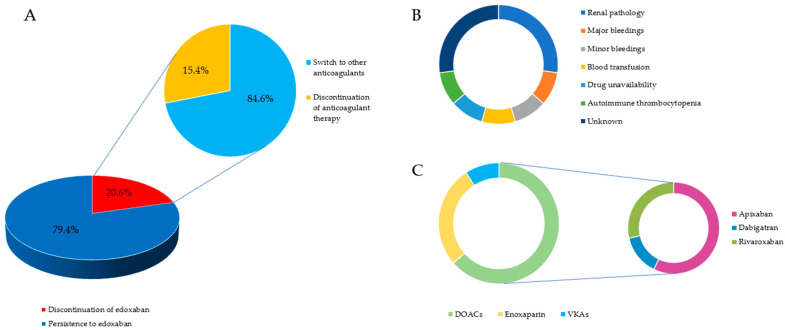
(**A**) A graphic description of the proportion of patients who persisted and those who did non persist on edoxaban therapy at long-term follow-up, reporting, within the second group, the proportion of patients who interrupted any kind of anticoagulant therapy and those who switched to other anticoagulants; (**B**) causes of switching to other anticoagulants and their frequency; (**C**) a list of anticoagulants chosen for switching to with the relative proportions.

**Figure 3 brainsci-13-01541-f003:**
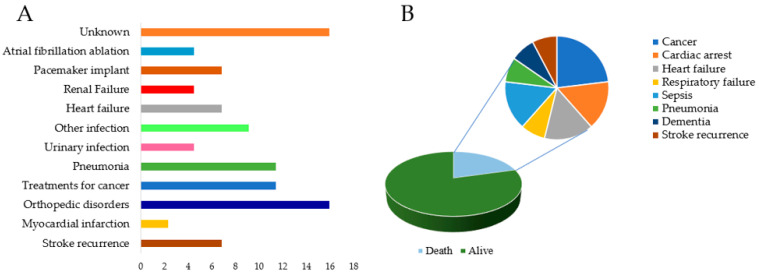
(**A**) A graphic representation of the various causes of hospitalizations after the index event and their frequencies; (**B**) a figure depicting the frequencies of the causes of the death in the study sample.

**Table 1 brainsci-13-01541-t001:** A descriptive table depicting several baseline clinical and demographic features of the study population, the median follow-up time, and the long-term degree of disability.

	Overall Study Population (n = 63)
Age at time of FUP (years)	81.0 (73.5–88.0)
Male sex	24 (38.1%)
FUP time (years)	4.6 (4.0–4.9)
mRS score at time of FUP	3.0 (0.0–4.0)

**Baseline stroke features and treatments**	
Wake-up onset	11 (17.5%)
NIHSS at admission	6.0 (3.0–11.0)
ASPECT score	10.0 (9.0–10.0)
Thrombolysis	25 (39.7%)
Mechanical Thrombectomy	16 (25.4%)

**Cerebrovascular risk factors**	
Smoking habit	12 (19.1%)
Diabetes	12 (19.1%)
Arterial Hypertension	44 (69.8%)
Dyslipidemia	15 (23.8%)
Previous Cerebrovascular disease	12 (19.0%)
Baseline CHADVASC score	5.0 (4.0–6.0)
Baseline HAS-BLED score	3.0 (3.0–3.0)

Categorical variables are expressed as number (%). Abbreviations: VKAs, vitamin K antagonists; DOACs, direct oral anticoagulants.

**Table 2 brainsci-13-01541-t002:** A descriptive table depicting the cause of failure to persist on edoxaban therapy and the type of medications chosen for the switching of anticoagulant therapy.

Edoxaban Non-Persistence	13 (20.6%)
Switch *	11 (84.6%)
Discontinuation *	2 (15.4%)
**Cause of switching to another anticoagulant ^†^**	
Renal pathology	3 (27.3%)
Major bleedings	1 (9.1%)
Minor bleedings	1 (9.1%)
Blood transfusion	1 (9.1%)
Drug unavailability	1 (9.1%)
Autoimmune thrombocytopenia	1 (9.1%)
Unknown	3 (27.3%)
**Anticoagulant chosen for switching ^†^**	
VKAs	1 (9.1%)
Enoxaparin	3 (27.3%)
DOACs	7 (63.6%)
Apixaban	4 (36.4%)
Dabigatran	1 (9.1%)
Rivaroxaban	2 (18.2%)
**Cause of drug discontinuation ^‡^**	
Unknown	2 (100%)

Categorical variables are expressed as number (%). * Proportion referring to 13 edoxaban non-adherent patients at long-term follow-up. ^†^ Percentage calculated considering the 11 patients who switched to another anticoagulant. ^‡^ Proportion based on the 2 patients who discontinued edoxaban therapy without switching to another anticoagulant. Abbreviations: VKAs, vitamin K antagonists; DOACs, direct oral anticoagulants.

**Table 3 brainsci-13-01541-t003:** Proportion of patients who changed to low-dose edoxaban therapy during the follow-up period and causes of dose reduction.

Low-Dose Edoxaban *	12 (24.0%)
Median age (years) ^⁋^	81.5 (74.8–88.0)
*Causes of reduction * ^⁋^	
Renal pathology	3 (25.0%)
Body weight ≤ 60 kg	8 (66.7%)
Unknown	1 (8.3%)

Categorical variables are expressed as number (%). * Proportion referring to the 50 patients adhering to edoxaban therapy at the time of follow-up. ^⁋^ Data referring to the 11 patients taking low-dose edoxaban.

**Table 4 brainsci-13-01541-t004:** A summary of the incidence of safety and efficacy outcomes of the study in the overall patient population.

		Overall Population (*n* = 63)
**Safety outcomes**		
n of patients with at least 1 hospitalization		34 (54.0%)
n of patients with ≥2 hospitalizations		10 (15.9%)
Total number of hospitalizations		44
	**Causes of hospitalization ***	
	Stroke recurrence	3 (6.8%)
	Myocardial Infarction	1 (2.3%)
	Orthopedic disorders	7 (15.9%)
	Surgery/Medical treatment for cancer	5 (11.4%)
	Pneumonia	5 (11.4%)
	Urinary infection	2 (4.5%)
	Other infections	4 (9.1%)
	Heart failure	3 (6.8%)
	Renal failure	2 (4.5%)
	Pacemaker implant	3 (6.8%)
	Atrial fibrillation ablation	2 (4.5%)
	Other causes	7 (15.9%)
**Minor bleedings**	Total	18 (28.6%)
Kind of minor bleedings	Bleeding gums	6 (9.5%)
	Epistaxis	1 (1.6%)
	Urinary bleeding	3 (4.8%)
	Fecal blood	4 (6.4%)
	Bruising	3 (4.8%)
**Major bleedings**		1 (1.6%)
Kind of major bleedings	Intracranial bleedings	0
	Bleedings requiring blood transfusion	1 (1.6%)
	Fatal bleeding	0
Blood transfusion		2 (3.2%)
**Efficacy outcomes**		
	Stroke recurrence	2 (3.2%)
	TIA	2 (3.2%)
	Myocardial infarction	1 (1.6%)
	Other thrombotic events	0
**Death**		13 (20.6%)
Causes of death ^⁋^		
	Cardiac arrest of unknown cause	2 (15.4%)
	Cancer	3 (23.1%)
	Heart failure	2 (15.4%)
	Respiratory failure	1 (7.7%)
	Sepsis	2 (15.4%)
	Pneumonia	1 (7.7%)
	Dementia	1 (7.7%)
	Stroke recurrence	1 (7.7%)

Categorical variables are expressed as number (%). * Proportions calculated considering the total number of hospitalizations (44 vs. 38). ^⁋^ Percentages referring to the 13 patients who died. Abbreviations: *n*, number; TIA, transient ischemic attack.

**Table 5 brainsci-13-01541-t005:** A statistical comparison of cardiovascular risk factors, comorbidities, and incidences of outcome events between patients assuming standard-dose edoxaban therapy and those taking 30 mg of edoxaban daily at the time of the long-term follow-up.

	Patients Persistent on Edoxaban (*n* = 50)	Standard-Dose Edoxaban (*n* = 38)	Low-Dose Edoxaban (*n* = 12)	*p*
Age at time of FUP (years)	81.0 (73.0–87.5)	80.0 (71.5–84.8)	86.5 (83.5–89.0)	0.010
Male sex	21 (42.0%)	19 (50.0%)	2 (16.7%)	*0.051*
FUP time (years)	4.5 (3.8–4.9)	4.6 (3.8–4.9)	4.2 (4.0–4.9)	0.724
mRS score at time of FUP	2.0 (0.0–4.0)	2.0 (0.0–4.0)	3.0 (0.8–4.5)	0.546
**Baseline stroke features and treatments**				
Wake-up onset	9 (18.0%)	7 (18.4%)	2 (16.7%)	1.000
NIHSS at admission	6.0 (3.0–11.0)	6.0 (3.0–11.8)	6.5 (3.8–10.3)	0.900
ASPECT score	10.0 (9.0–10.0)	10.0 (9.0–10.0)	10.0 (9.5–10.0)	0.848
Thrombolysis	20 (40.0%)	13 (34.2%)	7 (58.3%)	0.182
Mechanical Thrombectomy	15 (30.0%)	13 (34.2%)	2 (16.7%)	0.304
**Cerebrovascular risk factors**				
Smoking habit	10 (20.0%)	5 (13.2%)	5 (41.7%)	0.101
Diabetes	10 (20.0%)	7 (18.4%)	3 (25.0%)	0.690
Arterial Hypertension	34 (68.0%)	24 (63.2%)	10 (83.3%)	0.292
Dyslipidemia	12 (24.0%)	8 (21.1%)	4 (33.3%)	0.448
Previous Cerebrovascular disease	11 (22.0%)	8 (21.1%)	3 (25.0%)	1.000
**Long-term safety outcomes**				
Major bleedings	0 (0%)	0 (0%)	0 (0%)	
Intracranial bleedings	0 (0%)	0 (0%)	0 (0%)	
Minor bleedings	15 (30.0%)	13 (34.2%)	2 (16.7%)	0.304
Blood transfusion	0 (0%)	0 (0%)	0 (0%)	
**Long-term efficacy outcomes**				
Stroke recurrence	2 (4.0%)	1 (2.6%)	1 (8.3%)	0.426
TIA	2 (4.0%)	0 (0.0%)	2 (16.7%)	*0.054*
Myocardial Infarction	1 (2.0%)	1 (2.6%)	0 (0%)	1.000
Hospitalization	27 (54.0%)	21 (55.3%)	6 (50.0%)	1.000
Death from all causes	9 (18.0%)	6 (15.8%)	3 (25.0%)	0.668
Death from cardiovascular causes	3 (33.3%)	1 (2.6%)	2 (16.7%)	0.226

Categorical variables are expressed as number (%). Statistical comparisons were performed through a Mann–Whitney U-test for numerical variables and Fisher’s exact test for categorical ones. Significant findings are reported in bold, and in italic are the suggestive ones.

**Table 6 brainsci-13-01541-t006:** Comparisons between short- and long-term efficacy and safety data.

	SATES (Short-Term) *n* = 75	SATES (Long-Term) *n* = 63
Mean follow-up period (yrs)	0.25	4.4
Intracranial bleedings, *n* (annualized %)	0 (0)	0 (0)
Major bleedings, *n* (annualized %)	2 (10.7)	1 (0.4)
Minor bleedings, *n* (annualized %)	11 (58.7)	18 (6.4)
All thromboembolic events, *n* (annualized %)	0 (0)	6 (2.2)
Ischemic stroke, *n* (annualized %)	0 (0)	3 (1.1)
Transient ischemic attack, *n* (annualized %)	0 (0)	2 (0.7)
Acute myocardial infarction, *n* (annualized %)	0 (0)	1 (0.4)

Abbreviations: SATES, Prospective Observational Study of Safety of Early Treatment with edoxaban in Patients with Ischemic Stroke and Atrial Fibrillation.

## Data Availability

Data of the study are available upon reasonable request from the corresponding author.

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
