# Peer review of "Prolonged Secondary Stroke Prevention with Edoxaban: A Long-Term Follow-Up of the SATES Study"

_brainsci, 2023, doi:10.3390/brainsci13111541_

Round 1
Reviewer 1 Report
Comments and Suggestions for Authors
The text starts by describing the study as an observational, single-center study involving patients who had previously participated in the SATES Study. It provides information about the original SATES Study, which was non-randomized and aimed to assess the safety of early edoxaban treatment. The text outlines the criteria for including patients in this study. Patients were included if they provided informed consent for long-term follow-up. Exclusion criteria included refusal to participate and the inability to obtain recent information about the patient's clinical status and edoxaban intake. The text reports that only one patient (1.6%) experienced a major bleeding event, which was a massive epistaxis requiring a blood transfusion. No fatal bleeding or intracranial hemorrhage occurred. Five patients (7.9%) experienced thrombotic events during the follow-up period, with recurrent strokes, TIAs, and one myocardial infarction reported. Thirteen patients (20.6%) died during the study, with cancer being the most frequent cause of death. Overall, the study aimed to assess the safety and efficacy of edoxaban in patients with ischemic stroke and atrial fibrillation. Manuscript is well written. It provides detailed information on the outcomes, including safety and efficacy, and the reasons for hospitalization and death.
Author Response
We thank the reviewer for the comments.
Reviewer 2 Report
Comments and Suggestions for Authors
This is a well-built and interesting paper. The authors did an good job investigating long term adherence and influencing factors. However the result are presented in tables including multiple subpopulations which make it very difficult to follow, even with the legend. I strongly suggest to use graphs and figures to present the results. It will improve the quality and make it easier for the readers.
The results are mainly descriptive no statistical analysis was made, although relevant for the topic, authors should perform a formal statistical analysis that can provide insights of factors affecting adherence. Additionally, their claims of effectiveness and safety on their conclusion are not supported by a formal survival analysis, this will improve the significance of the research.
Comments on the Quality of English Languagenone
Author Response
We thank the reviewer for the suggestion. We added two figures (Figure 2 and Figure 3), composed by graphics depicting the results of the study in order to simplify the comprehensiveness of the manuscript
- Additionally, their claims of effectiveness and safety on their conclusion are not supported by a formal survival analysis, this will improve the significance of the research.
As suggested by the reviewer we modified the manuscript in several points:
- Study title: We removed “is safe and effective”, changing the title as follows: “Prolonged secondary stroke prevention with edoxaban: A long-term follow-up of the SATES study”
- Abstract: “Edoxaban seems to be an effective and safe long-term treatment. The persistence rate of edoxaban therapy is optimal after more than three years of treatment.”
- Discussion: “In this study, we found that edoxaban therapy seems to be safe and effective in the long-term (> 3 years of follow-up).”
- Conclusions: “Our study suggests that edoxaban is seems to be an effective, safe, and well-tolerated drug in elderly patients with NVAF and previous ischemic stroke. However, due to many limitations, further prospective, multicenter, larger-sample studies are needed to strengthen the evidence emerged from our study.”
Reviewer 3 Report
Comments and Suggestions for Authors
This study assessed the long-term safety and efficacy of edoxaban. Its primary goal was to examine its safety by evaluating major bleeding incidents, followed by an assessment of thromboembolic events and the continued use of edoxaban over an extended period. The study included 63 ischemic stroke patients with a mean follow-up of 4.4 years. Only one patient experienced a major bleeding event, while six thromboembolic events occurred in five patients. The study found that long-term edoxaban therapy is effective and safe, with an optimal persistence rate of over three years of treatment, highlighting its potential for long-term use. Overall, the study is well-designed and the finding is important.
1. Please revise the first raw to table 2 to describe the total number of patients with non-adherence (n = 13).
2. If possible, please compared the outcome of patients receiving edoxaban with and without dose adjustment.
3. English editing is needed.
Comments on the Quality of English LanguagePlease corrected some typo
Author Response
Please revise the first raw to table 2 to describe the total number of patients with non-adherence (n = 13).
- We thank the reviewer for the suggestion. We revised the first raw of Table 2, reporting the total number of non-persistent patients.
If possible, please compared the outcome of patients receiving edoxaban with and without dose adjustment.
- We thank the reviewer for the suggestion. We performed a comparison of risk factors and outcomes between patients taking standard dose and those taking low-dose edoxaban and modified the text as follows:
“In statistical analysis:
Comparisons among groups were performed through Mann-Whitney U-test or Fisher’s exact test, as appropriate. Statistical significance was settled for two-tailed p<0.05.
In results:
3.6 Comparisons between low-dose and standard-dose edoxaban therapy
Considering the 50 patients who maintained edoxaban therapy at the time of the long-term follow-up, we performed a statistical comparison between the group of subjects taking edoxaban at standard dose (60 mg) and those assuming low-dose edoxaban therapy (30 mg).
We observed that patients taking low-dose edoxaban presented a higher median age than those assuming edoxaban 60 mg [Standard-dose: 80.0 (71.5 – 84.8) years vs Low-dose: 86.5 (83.5-89.0) years; p = 0.010]. Female patients tended to assume more frequently a low dose of edoxaban than men [Standard-dose: 19 (50%) vs Low-dose: 10 (83.3%); p = 0.051].
Concerning the analysis of outcome events, we found that only TIAs tended to be more frequent in the low-dose group, without reaching statistical significance [Standard-dose: 0 (0%) vs Low-dose: 2 (16.7%); p = 0.054]. All the other outcome events did not differ significantly between groups. For details, please refer to Table 5.
Table 5. A statistical comparison of cardiovascular risk factors, comorbidities, and incidence of outcome events between patients assuming standard-dose edoxaban therapy and those taking 30 mg edoxaban daily at the time of the long-term follow-up.
Patients persistent to edoxaban (n=50) |
Standard-dose edoxaban (n=38) |
Low-dose edoxaban (n=12) |
p
|
||
Age at time of FUP (years) |
81.0 (73.0 – 87.5) |
80.0 (71.5 - 84.8) |
86.5 (83.5 - 89.0) |
0.010 |
|
Male sex |
21 (42.0%) |
19 (50.0%) |
2 (16.7%) |
0.051 |
|
FUP time (years) |
4.5 (3.8 - 4.9) |
4.6 (3.8 - 4.9) |
4.2 (4.0 – 4.9) |
0.724 |
|
mRS score at time of FUP |
2.0 (0.0 - 4.0) |
2.0 (0.0 – 4.0) |
3.0 (0.8 – 4.5) |
0.546 |
|
Baseline stroke features and treatments |
|||||
Wake-up onset |
9 (18.0%) |
7 (18.4%) |
2 (16.7%) |
1.000 |
|
NIHSS at admission |
6.0 (3.0 - 11.0) |
6.0 (3.0 - 11.8) |
6.5 (3.8 – 10.3) |
0.900 |
|
ASPECT score |
10.0 (9.0 - 10.0) |
10.0 (9.0 - 10.0) |
10.0 (9.5 - 10.0) |
0.848 |
|
Thrombolysis |
20 (40.0%) |
13 (34.2%) |
7 (58.3%) |
0.182 |
|
Mechanical Thrombectomy |
15 (30.0%) |
13 (34.2%) |
2 (16.7%) |
0.304 |
|
Cerebrovascular risk factors |
|||||
Smoking habit |
10 (20.0%) |
5 (13.2%) |
5 (41.7%) |
0.101 |
|
Diabetes |
10 (20.0%) |
7 (18.4%) |
3 (25.0%) |
0.690 |
|
Arterial Hypertension |
34 (68.0%) |
24 (63.2%) |
10 (83.3%) |
0.292 |
|
Dyslipidemia |
12 (24.0%) |
8 (21.1%) |
4 (33.3%) |
0.448 |
|
Previous Cerebrovascular disease |
11 (22.0%) |
8 (21.1%) |
3 (25.0%) |
1.000 |
|
Long-term safety Outcomes |
|||||
Major bleedings |
0 (0%) |
0 (0%) |
0 (0%) |
||
Intracranial bleedings |
0 (0%) |
0 (0%) |
0 (0%) |
||
Minor bleedings |
15 (30.0%) |
13 (34.2%) |
2 (16.7%) |
0.304 |
|
Blood transfusion |
0 (0%) |
0 (0%) |
0 (0%) |
||
Long-term efficacy outcomes |
|||||
Stroke recurrence |
2 (4.0%) |
1 (2.6%) |
1 (8.3%) |
0.426 |
|
TIA |
2 (4.0%) |
0 (0.0%) |
2 (16.7%) |
0.054 |
|
Myocardial Infarction |
1 (2.0%) |
1 (2.6%) |
0 (0%) |
1.000 |
|
Hospitalization |
27 (54.0%) |
21 (55.3%) |
6 (50.0%) |
1.000 |
|
Death for all causes |
9 (18.0%) |
6 (15.8%) |
3 (25.0%) |
|
0.668 |
Death for cardiovascular causes |
3 (33.3%) |
1 (2.6%) |
2 (16.7%) |
|
0.226 |
Statistical comparisons were performed through Mann-Whitney U-test for numerical variables and Fisher’s exact test for categorical ones. Significant findings are reported in bold and in italic the suggestive ones
English editing is needed.
- We than the reviewer for the suggestion. We have tried our best to improve the English editing.
Reviewer 4 Report
Comments and Suggestions for Authors
The authors of the manuscript, ‘Prolonged secondary stroke prevention with edoxaban is safe and effective: A long term follow-up of the SATES study' have tried to examine the long-term efficacy and safety of edoxaban in ischemic stroke patients by probing major bleeding incidents, thromboembolic events and persistence of edoxaban therapy as safety indicators. They collected data telephonically by a structured questionnaire. The paper has some major drawbacks in design and execution. Authors should clarify/rectify some of the following points for better presentation and confirmation.
1. Many studies this year have investigated this aspect (PMIDs: 37711730, 36817122, 37282378) and some reports have examined this perspective in previous years (PMIDs: 26019695, 27207971, 244251359). Authors may argue that its long-term efficacy remains to be investigated, but authors did not compare their long-term follow-up evaluation with short-term follow-up evaluation, which according to me are similar. Authors should clarify that why long-term therapy is better than short-term edoxaban therapy in secondary stroke prevention by comparison. Without comparison it is vague.
2. Why authors could not use telemedicine which would have been better in collecting the relevant information than using telephone. The authors should also explain that how their study prevented or minimized false positivity and false negativity in such type of information collected.
3. The data of 63 patients is very small to reach any conclusion. The authors should have analyzed the statistical power of the study. Sample size showing statistical power above 80% could reject the null and is generally considered to give unambiguous results
4. Authors have used a very uncommon style of writing the title of the tables. Titles should be small and the rest information can be put as footnotes.
5. Authors should explain that why after discontinuation of edoxaban by 5% of the patients for safety outcomes (although authors have written that, 'the reason of the discontinuation was not clarified'-Line 170) relates to the conclusion that edoxaban is safe. Authors should explain that, is 28.6% cases of minor bleeding in edoxaban therapy is normal to ascribe this drug as safe and effective.
6. The term, ‘ultraoctagenarians' used by authors is new. What does it stand for, otherwise individuals with ages from 80-90 years or 80-89 years are called octogenarian by Merriam-Webster, Cambridge, Collins dictionaries, and even by Britannica. From where, the authors have taken this nomenclature? Authors should remove this word from the whole text.
7. Authors must compare results deduced from this long-term follow-up of edoxaban therapy with reports of short-term follow-up to confirm its efficacy.
8. If authors claim that it has not been tested in the elderly population (octogenarians) then why authors didn't investigate its efficacy by adjusting the effect of age as a confounder?
9. Apropos to three major limitations i.e. small sample size, collection of information telephonically, and no comparison with short-term follow-up, the title depicting ‘Prolonged secondary stroke prevention with edoxaban is safe and effective’ seems to be a sweeping statement rather than suggestive disclosure.
Comments on the Quality of English LanguageMinor English language editings are required
Author Response
Many studies this year have investigated this aspect (PMIDs: 37711730, 36817122, 37282378) and some reports have examined this perspective in previous years (PMIDs: 26019695, 27207971, 244251359). Authors may argue that its long-term efficacy remains to be investigated, but authors did not compare their long-term follow-up evaluation with short-term follow-up evaluation, which according to me are similar. Authors should clarify that why long-term therapy is better than short-term edoxaban therapy in secondary stroke prevention by comparison. Without comparison it is vague.
- We thank the reviewer for the suggestion. Unfortunately, we decided to include in the discussion only original articles, excluding reviews and case series. We have however considered in the manuscript some of the articles suggested by the reviewer (PMID: 36817122, 27207971, 24251359), in order to compare our efficacy and safety data with those emerged from previous literature. Furthermore, as suggested by the reviewer, we performed a comparison of the annualized rate of short-term versus long-term efficacy and safety outcomes of edoxaban therapy in our patient population. In particular, we compared the following safety and efficacy parameters: intracranial bleeding, major bleeding, minor bleeding, ischemic stroke, acute myocardial infarction. So, we modified the text as follows:
3.7 Comparisons between short and long-term efficacy and safety data
Comparing the efficacy and safety data from the 3 months follow-up of our previous study [10] and the long-term follow-up in our cohort of stroke patients we observed a higher annualized rate of both major (annualized rate of short-term vs long term follow-up: 10.7% vs 0.4%) and minor bleedings (annualized rate of short-term vs long term follow-up: 58.7% vs 6.4%) during the short-term period than the long-term one. On the other hand, the annualized rate of thromboembolic events was slightly higher in the long-term follow-up cohort of patients. For details, please refer to Table 6.
Table 6. Comparisons between short and long-term efficacy and safety data
|
SATES (short-term) n = 75 |
SATES (long-term) n = 63 |
Mean follow-up period (yrs) |
0.25 |
4.4 |
Intracranial bleedings, n (annualized %) |
0 (0) |
0 (0) |
Major bleedings, n (annualized %) |
2 (10.7) |
1 (0.4) |
Minor bleedings, n (annualized %) |
11 (58.7) |
18 (6.4) |
All thromboembolic events, n (annualized %) |
0 (0) |
6 (2.2) |
Ischemic stroke, n (annualized %) |
0 (0) |
3 (1.1) |
Transient ischemic attack, n (annualized %) |
0 (0) |
2 (0.7) |
Acute myocardial infarction, n (annualized %) |
0 (0) |
1 (0.4) |
Abbreviations: SATES, Prospective Observational Study of Safety of Early Treatment with Edoxaban in Patients with Ischemic Stroke and Atrial Fibrillation
Why authors could not use telemedicine which would have been better in collecting the relevant information than using telephone. The authors should also explain that how their study prevented or minimized false positivity and false negativity in such type of information collected.
- We thank the reviewer for the suggestion. Unfortunately, in this study we only used a structured telephone interview with the patient, family members, and caregivers, also due to its ease of use for patients. In any case, we tried to collect only information that can be easily collected by telephone. We have included this observation in the limitations as follows "… The collection of long-term follow-up data through a telephone interview, as patients or their legal representatives were not always able to accurately report subjects’ clinical information, leading to missing data…”
The data of 63 patients is very small to reach any conclusion. The authors should have analyzed the statistical power of the study. Sample size showing statistical power above 80% could reject the null and is generally considered to give unambiguous results.
- Given the exploratory and descriptive nature of this study, no sample size analysis can be performed. So, we modified the text in statistical analysis as follows: “Given the exploratory and descriptive nature of this study, no sample size analysis was performed. Gaussian distribution of quantitative …”.
Authors have used a very uncommon style of writing the title of the tables. Titles should be small and the rest information can be put as footnotes.
- We thank the reviewer for the suggestion. We modified Tables titles and footnotes throughout the whole manuscript, as suggested.
Authors should explain that why after discontinuation of edoxaban by 5% of the patients for safety outcomes (although authors have written that, 'the reason of the discontinuation was not clarified'-Line 170) relates to the conclusion that edoxaban is safe. Authors should explain that, is 28.6% cases of minor bleeding in edoxaban therapy is normal to ascribe this drug as safe and effective.
- We thank the reviewer for the suggestion. We reported, for clarity, the annualized rate of minor bleedings in the results section, modifying the result section as follows: “Therefore, the annualized rate of minor bleedings in our edoxaban-taking population was 6.4%”. We also compared such a result with evidence emerged from previous literature, modifying the discussion as follows: “The incidence of minor bleedings in our patient population (annualized rate of 6.4%), falls within the range observed in the literature, ranging from an annualized rate of 3.52% (for the low-dose) [17] to 7.23% [20]”.
Furthermore, we clarified the results in Line 176, as follows: “For the two patients who interrupted anticoagulant therapy, the reason of this discontinuation was not clarified, but it was not attributable to the occurrence of an outcome event, as reported by the patients their caregivers.”.
The term, ‘ultraoctagenarians' used by authors is new. What does it stand for, otherwise individuals with ages from 80-90 years or 80-89 years are called octogenarian by Merriam-Webster, Cambridge, Collins dictionaries, and even by Britannica. From where, the authors have taken this nomenclature? Authors should remove this word from the whole text.
- We thank the reviewer for the suggestion, and we apologize for the mistake. We changed the term “ultraoctagenarians” in “octogenarians” in the whole manuscript.
Authors must compare results deduced from this long-term follow-up of edoxaban therapy with reports of short-term follow-up to confirm its efficacy.
- We thank the reviewer for the suggestion. In the manuscript, we considered the annualized rate of both safety and efficacy outcomes, trying to reduce the impact of the follow-up time in the incidence of such outcome events. However, as mentioned before, we performed a comparison between the short-term and the long-term follow-up of our patient population. We also compared our results with some of the reports suggested by the reviewer (PMID: 36817122, 27207971, 24251359), in order to compare our results with those emerging from previous literature and we added the following sentence to the manuscript: “Considering the incidence of minor bleedings in our patient population (annualized rate of 6.4%), we can see that it is comprised in that reported in previous literature, ranging from an annualized rate of 3.52% (for the low-dose) [17] to 7.23% [20]”.
If authors claim that it has not been tested in the elderly population (octogenarians) then why authors didn't investigate its efficacy by adjusting the effect of age as a confounder?
- We thank the reviewer for the suggestion. Unfortunately, due to the small sample size of our study, no formal statistical analyses have been performed among groups. Consequently, no multivariate logistic regressions corrected for confounding factors (as age) could be performed in this study. It must however be considered that the median age of our patient population was 81.5 (74.8 – 88.0) years, and that the number of patients younger of 75 years were only 17 (27.0%) and those younger than 70 years were only 10 (15.9%). So, we can consider that our population was mainly composed by elderly patients.
Apropos to three major limitations i.e. small sample size, collection of information telephonically, and no comparison with short-term follow-up, the title depicting ‘Prolonged secondary stroke prevention with edoxaban is safe and effective’ seems to be a sweeping statement rather than suggestive disclosure.
- As suggested by the reviewer we modified the manuscript in several points:
- Title: removing “is safe and effective”: Prolonged secondary stroke prevention with edoxaban: A long-term follow-up of the SATES study”
- Abstract: “Edoxaban seems to be an effective and safe long-term treatment. The persistence rate of edoxaban therapy is optimal after more than three years of treatment.”
- Discussion: “In this study, we found that edoxaban therapy seems to be safe and effective in the long-term (> 3 years of follow-up).”
- Conclusions: “Our study suggests that edoxaban is seems to be an effective, safe, and well-tolerated drug in elderly patients with NVAF and previous ischemic stroke. However, due to many limitations, further prospective, multicenter, larger-sample studies are needed to strengthen the evidence emerged from our study.”
Round 2
Reviewer 2 Report
Comments and Suggestions for Authors
I thank the authors for their corrections. The relevance of the paper increased significantly.
A final suggestion, the radar chart in figure 3 might not be the best to represent the frequency of occurrence, I suggest considering using a bar plot
Comments on the Quality of English LanguageNA
Author Response
Reviewer 2
A final suggestion, the radar chart in figure 3 might not be the best to represent the frequency of occurrence, I suggest considering using a bar plot
Thanks for your suggestions. We modified the figure as follows:
Reviewer 4 Report
Comments and Suggestions for Authors
Authors in their revision have sidelined the pointers raised by me. I suggest authors to rectify it rather than simply admit it.
I am reminding the following unsatisfied points to authors once again.
The authors should also explain that how their study prevented or minimized false positivity and false negativity in such type of information collected.
This point I sidelined without addressing the core issue
The data of 63 patients is very small to reach any conclusion. The authors should have analyzed the statistical power of the study. Sample size showing statistical power above 80% could reject the null and is generally considered to give unambiguous results
I am not at all convinced that descriptive studies do not need appropriate power and sample size for convincing results. If going by simple definitions then it is also mandatory that descriptive studies should have 10 to 20% representation of the population. The authors did not address this.
Authors should explain that why after discontinuation of edoxaban by 5% of the patients for safety outcomes (although authors have written that, 'the reason of the discontinuation was not clarified'-Line 170) relates to the conclusion that edoxaban is safe. Authors should explain that, is 28.6% cases of minor bleeding in edoxaban therapy is normal to ascribe this drug as safe and effective.
Why authors sidelined their inference of 28.6% of minor bleeding.
Apropos to three major limitations i.e. small sample size, collection of information telephonically, and no comparison with short-term follow-up, the title depicting ‘Prolonged secondary stroke prevention with edoxaban is safe and effective’ seems to be a sweeping statement rather than suggestive disclosure.
Why no defense has been put-froth against this?
Comments on the Quality of English LanguageExtensive English language editing is required.
Author Response
- The authors should also explain that how their study prevented or minimized false positivity and false negativity in such type of information collected. This point I sidelined without addressing the core issue.
I apologize for not responding to this issue upon first review.
To prevent and minimize false positivity and false negativity in such type of information collected we have adopted the following strategies:
- we used a uniform CRF for all telephone interviews;
- the "death" data was confirmed in terms of date and diagnosis by the national register of the Ministry of Health ( https://iam.regione.lazio.it/authenticationendpoint/login.do?commonAuthCallerPath=%2Fsamlsso&forceAuth=false&passiveAuth=false&tenantDomain=carbon.super&sessionDataKey=52a60045-04e0-40f0-9949-770cdf02cd28&relyingParty=it.laziocrea.sismed.sp&type=samlsso&sp=Sismed&isSaaSApp=false&authenticators=SAMLSSOAuthenticator%3ASHARED_lepidaid%3ASHARED_intesaid%3ASHARED_cie%3ASHARED_infocertid%3ASHARED_namirialid%3ASHARED_arubaid%3ASHARED_spiditalia%3ASHARED_sielteid%3ASHARED_posteid%3ASHARED_teamsystem%3ASHARED_timid%3ASHARED_etnaid%3ASHARED_infocamereid%3BBasicAuthenticator%3ALOCAL%3ALOCAL
- To minimize false positives, the data relating to the primary safety endpoints were confirmed by sending discharge letters and/or admissions to the ED via email by patients and/or carers. In this regard, due to the role of our hospital as a hub for ischemic cerebrovascular diseases and brain hemorrhages, 71% of patients' hospitalizations and/or ED visits occurred at our center, for which we had the data available. In any case, even in this case we performed a telephone contact at the time of recruitment to evaluate if other events had occurred in the subsequent time interval.
- The acquisition of CRF data was confirmed by both the patient and the caregiver
- The data of 63 patients is very small to reach any conclusion. The authors should have analyzed the statistical power of the study. Sample size showing statistical power above 80% could reject the null and is generally considered to give unambiguous results.
I am not at all convinced that descriptive studies do not need appropriate power and sample size for convincing results. If going by simple definitions then it is also mandatory that descriptive studies should have 10 to 20% representation of the population. The authors did not address this.
- We estimated the sample size for this cross-sectional study considering the occurrence of major bleedings as primary endpoint. The expected prevalence of major bleedings in a population taking direct anticoagulants can be found in literature [17] and is 2.75%.
We thus applied the following formula:
Where:
at 95% Confidence Interval or 5% level of significance (type-I error) = 1.96
p is the expected prevalence = 0.0275
q = (1-p) = 0.9725
d is the margin of error or precision = 0.05
And the sample size (n) resulted = 41 patients.
So we modified the text as follows:
“The sample size of this cross-sectional study was estimated considering the occurrence of major bleedings as the primary endpoint, and considering an expected annualized rate of 2.75% major bleedings in a DOACs-taking population, as previously reported [17]. Based and these premises and considering a confidence level of 95% and a precision of the estimate of 5%, the minimum number of patients to be included in the study was 41 patients.”
- Giugliano, R.P.; Ruff, C.T.; Braunwald, E.; Murphy, S.A.; Wiviott, S.D.; Halperin, J.L.; Waldo, A.L.; Ezekowitz, M.D.; Weitz, J.I.; Spinar, J.; et al. Edoxaban versus warfarin in patients with atrial fibrillation. N Engl J Med 2013, 369, 2093-2104, doi:10.1056/NEJMoa1310907
- Authors should explain that why after discontinuation of edoxaban by 5% of the patients for safety outcomes (although authors have written that, 'the reason of the discontinuation was not clarified'-Line 170) relates to the conclusion that edoxaban is safe. Authors should explain that, is 28.6% cases of minor bleeding in edoxaban therapy is normal to ascribe this drug as safe and effective. Why authors sidelined their inference of 28.6% of minor bleeding.
- As reported in the revised version both in the text and in the new table 6 we added the annualized minor bleeding data (6.4%) which is a data comparable with the literature and so we did.
As reported in the discussion, we compared our annualized data with the annualized data of 2 of the studies that you suggested in the previous review: "The incidence of minor bleedings in our patient population (annualized rate of 6.4%), falls within the range observed in the literature, ranging from an annualized rate of 3.52% (for the low-dose) [17] to 7.23% [20].
- Giugliano, R.P.; Ruff, C.T.; Braunwald, E.; Murphy, S.A.; Wiviott, S.D.; Halperin, J.L.; Waldo, A.L.; Ezekowitz, M.D.; Weitz, J.I.; Spinar, J.; et al. Edoxaban versus warfarin in patients with atrial fibrillation. N Engl J Med 2013, 369, 2093-2104, doi:10.1056/NEJMoa1310907.
- Grymonprez, M.; De Backer, T.L.; Bertels, X.; Steurbaut, S.; Lahousse, L. Long-term comparative effectiveness and safety of dabigatran, rivaroxaban, apixaban and edoxaban in patients with atrial fibrillation: A nationwide cohort study. Front Pharmacol 2023, 14, 1125576, doi:10.3389/fphar.2023.1125576."
- Apropos to three major limitations i.e. small sample size, collection of information telephonically, and no comparison with short-term follow-up, the title depicting ‘Prolonged secondary stroke prevention with edoxaban is safe and effective’ seems to be a sweeping statement rather than suggestive disclosure. Why no defense has been put-froth against this?
- As reviewer have seen, we have changed the title and the text because, as suggested by the reviewer, being a descriptive study and not having a control group it would have been too speculative.